# IllusionVQA: A Challenging Optical Illusion Dataset for Vision Language Models

**Haz Sameen Shahgir**[1,3][*] **Khondker Salman Sayeed**[1][*] **Abhik Bhattacharjee**[1],
**Wasi Uddin Ahmad**[2], **Yue Dong**[3], **Rifat Shahriyar**[1]

Bangladesh University of Engineering and Technology[1],
AWS AI Labs[2][†] University of California Riverside[3]
sameen2080@gmail.com, salkhon050@gmail.com

## Abstract

The advent of Vision Language Models (VLM) has allowed researchers to investigate the visual understanding of a neural network using natural language. Beyond object classification and detection, VLMs are capable of visual comprehension and common-sense reasoning. This naturally led to the question: How do VLMs respond when the image itself is inherently *unreasonable*? To this end, we present IllusionVQA: a diverse dataset of challenging optical illusions and hard-to-interpret scenes to test the capability of VLMs in two distinct multiple-choice VQA tasks - comprehension and soft localization. GPT4V, the best performing VLM, achieves 62.99% accuracy (4-shot) on the comprehension task and 49.7% on the localization task (4-shot and Chain-of-Thought). Human evaluation reveals that humans achieve 91.03% and 100% accuracy in comprehension and localization. We discover that In-Context Learning (ICL) and Chain-of-Thought reasoning substantially degrade the performance of Gemini-Pro in the localization task. Tangentially, we discover a potential weakness in the ICL capabilities of VLMs: they fail to locate optical illusions even when the correct answer is in the context window as a few-shot example.[1]

## 1  Introduction

Optical illusions are characterized by a visual percept that arguably differs from reality. Illusions come in a wide variety; their categorization is difficult because the underlying cause is often unclear (Bach & Poloschek, 2006; Gregory, 1997a). Figure 2 shows examples that are distinct from each other but all fall under the umbrella of optical illusions. Figure 2a shows an impossible object where - *"Parts of the object (in this case the elephant's legs) become the background, and vice versa."* (Shepard, 1990). Figure 2b depicts a 3-by-3 grid where some cells appear pale-yellow, but upon closer inspection, it is evident that all cells are perfectly white. Figure 2c shows a realistic 3D drawing of a white cube with a cup positioned as resting as if on top of the fictional cube. Although all three examples are arguably optical illusions, they have little in common. Despite the challenges in optical illusion classification, Gregory (1997b) proposed four main classes based on appearance - ambiguities, distortions, paradoxes, and fictions. Based on this framework, Figure 2a is a *paradox* since the object depicted cannot truly exist. Figure 2b is a *distortion* since none of the regions is pale yellow, although it might appear so. Figure 2c is a Trompe-l'œil illustration and its classification as an optical illusion is contentious (Wade & Hughes, 1999), although it might fall under the aforementioned *fiction* category.

The cause of our perception of optical illusions is a rich field of research in Cognitive Psychology (Helmholtz, 1948; Gregory, 1968; Gentilucci et al., 1996). Motivated by similarities

---

[*] Equal Contribution
[†] The work does not relate to the author's position at Amazon.
[1] The code and datasets are available at https://github.com/csebuetnlp/IllusionVQA.

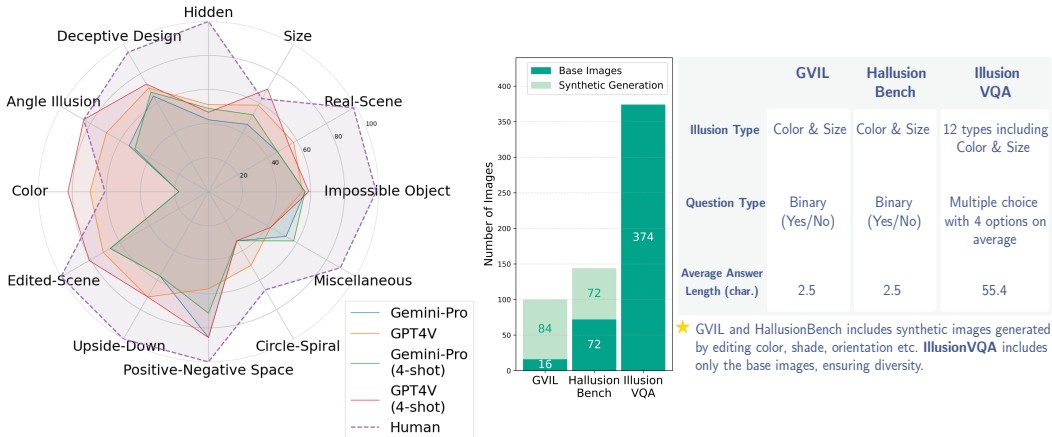

Figure 1: Left: Comparison of human and VLM performance in IllusionVQA-Comprehension. Right: Comparison of IllusionVQA with prior illusion datasets - GVIL (Zhang et al., 2023) and HallusionBench (Liu et al., 2023a).

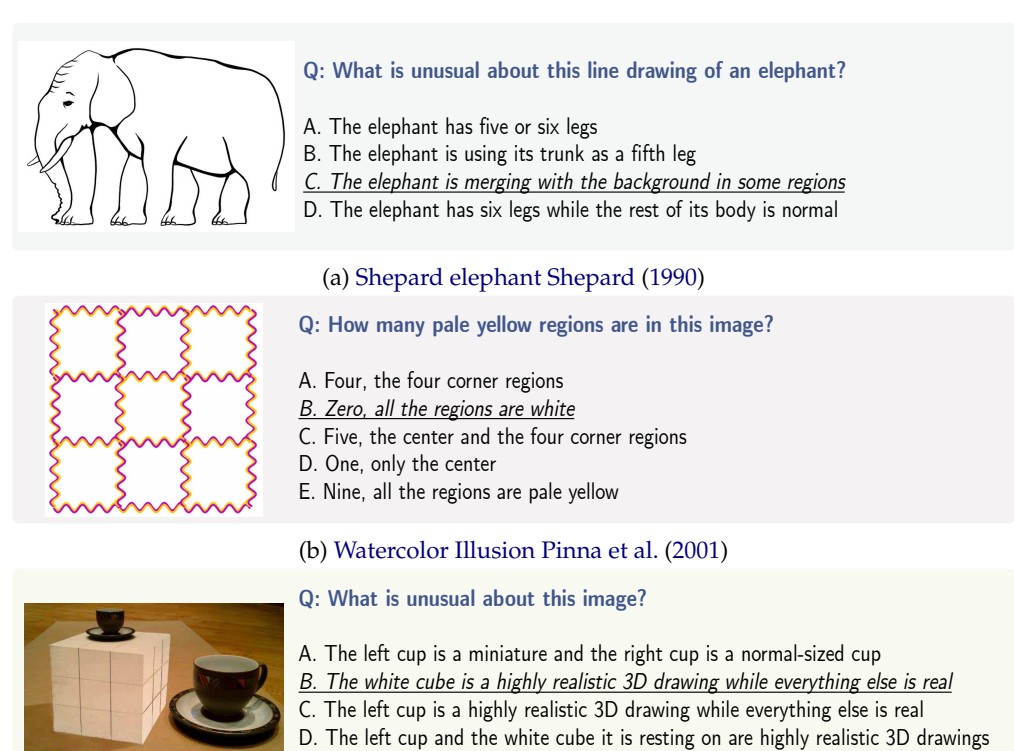

(a) Shepard elephant Shepard (1990)

(b) Watercolor Illusion Pinna et al. (2001)

(c) A trompe l'oeil art, not traditionally considered as an optical illusion in the cognitive sciences.

Figure 2: Examples of optical illusions in IllusionVQA-Comprehension

in human visual cognition and artificial neural networks (Cichy et al., 2016; Eickenberg et al., 2017), researchers have explored if and how artificial neural networks perceive optical illusions (Gomez-Villa et al., 2019; 2020; Sun & Dekel, 2021; Lonnqvist et al., 2021; Hirsch & Tal, 2020). However, the scope and generalizability of these works were limited by the need for a case-by-case analysis of the activations and gradients of the model in response to individual optical illusions. As a remedy, Zhang et al. (2023) was the first to explore optical illusions through small VLMs (up to 13 billion parameters) and to investigate their internal beliefs about optical illusions through natural language.

Unlike prior work, we curate challenging optical illusions from the Internet that span 12 distinct categories inherited from cognitive psychology studies. We craft detailed question-answer pairs and several incorrect options designed to probe the abilities of VLMs. We frame the problem as a standard visual question-answering task (VQA) and evaluate state-of-the-art VLMs such as GPT4V (Achiam et al., 2023) and Gemini-Pro (Team et al., 2023) in addition to smaller open-source models. Our detailed categorization and extensive human evaluation allow for a fine-grained comparison of human and VLM cognition of optical illusions along different axes. Our contributions are summarized as follows.

1. We introduce IllusionVQA, a novel dataset designed to rigorously test the ability of VLMs to locate and comprehend challenging optical illusions.
2. We comprehensively test a wide range of open-source and closed-source VLMs. Our experiments reveal that GPT4V is the most capable model on illusion comprehension and localization but still lags significantly behind human-level performance.
3. Through experiments, we discover that state-of-the-art VLMs can locate ordinary objects accurately, but struggle with optical illusions. Furthermore, these models exhibit inconsistencies when evaluated through In-Context Learning and Chain-of-Thought reasoning.

## 2 Related Work

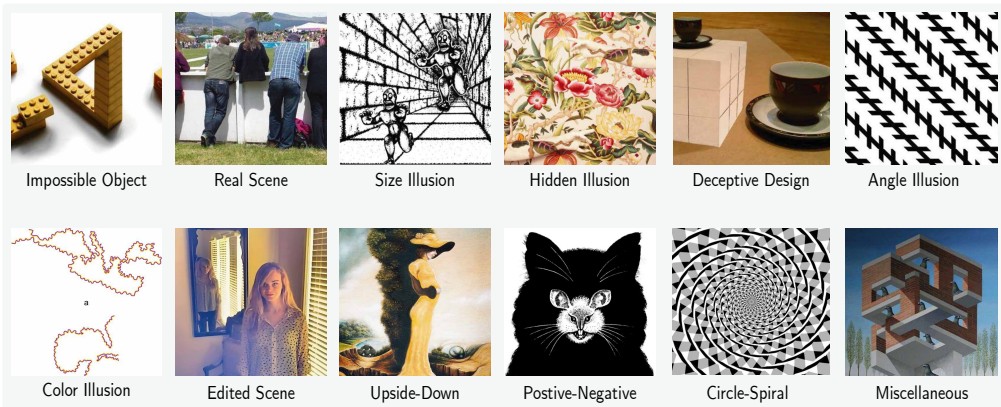

Figure 3: Categories in IllusionVQA-Comprehension. Refer to Appendix D for details.

Researchers have shown that convolutional neural networks trained on ordinary images are susceptible to certain optical illusions, similar to humans (Gomez-Villa et al., 2019; 2020; Afifi & Brown, 2019; Benjamin et al., 2019; Sun & Dekel, 2021). However, most previous experiments have focused on specific categories of optical illusions and their methodologies do not generalize to all types.

Zhang et al. (2023) is the first to probe how VLMs perceive optical illusions through natural language. The authors collected 16 root images of optical illusions and created 100 variants through manual editing. They ask the model a Yes/No question to test whether a VLM has been "fooled" by an optical illusion. They tested multiple open-source VLMs (up to 13B parameters) and discovered that larger VLMs are more susceptible to optical illusions. Liu et al. (2023a) introduces HallusionBench, a benchmark for testing VLM susceptibility to hallucination and optical illusions which includes 72 base images of optical illusions mostly derived from Zhang et al. (2023). Bitton-Guetta et al. (2023) tests VLM comprehension on *weird* images, e.g. *"A lit candle inside a sealed bottle"*. They generate 500 such images using Text-to-Image generative models (Ramesh et al., 2021; Rombach et al., 2022) and employ human annotators to create descriptions of why the image is *weird*. They found that GPT3 with oracle image description achieves 68% accuracy, while humans achieve 95%.

Prior research has yielded multiple tools for synthetic optical illusion generation. Hirsch & Tal (2020) and Fan & Zeng (2023) automatically produce color-related and grating-related optical illusions, respectively. Gomez-Villa et al. (2022) explores creating illusions using

GANs. The primary limitation of synthetic illusion generation is the limited variety in the illusions they can create, especially compared to the wide spectrum of optical illusions discovered in the literature. As such, relying solely on existing synthetic illusion generation algorithms to evaluate VLMs severely limits the scope of the evaluation.

In contrast to prior works, we collect images of optical illusions from the internet to ensure data diversity and test VLMs along multiple axes. We manually create a multiple-choice question-answering (QA) dataset where each question is crafted to have only one unambiguous answer while all other options appear plausible but are incorrect. Beyond comprehension, we test how well VLMs can locate geometrically impossible objects in an image. Our experiments probe VLM's dependence on language priors (Goyal et al., 2017) and feature entanglement (Tang et al., 2023) in the context of optical illusions.

## 3 The IllusionVQA Dataset

IllusionVQA is a Visual Question Answering (VQA) dataset with two sub-tasks. The first task tests comprehension on 435 instances in 12 optical illusion categories. Each instance consists of an image with an optical illusion, a question, and 3 to 6 options, one of which is the correct answer. We refer to this task as **IllusionVQA-Comprehension**. The second task tests how well VLMs can differentiate geometrically impossible objects from ordinary objects when two objects are presented side by side. The task consists of 1000 instances following a similar format to the first task. We refer to this task as **IllusionVQA-Soft-Localization**. This section details image collection and filtering, followed by QA generation and classification for the Comprehension task and procedural QA generation for the Soft-Localization task.

### 3.1 Image Collection and Filtering

We scraped more than 3500 images of optical illusions from multiple online repositories. We manually inspected each image to ensure that they were optical illusions. Since VLMs are trained on web-scraped data, dataset contamination is a major concern for web-scraped image datasets. Liu et al. (2023a) notes that GPT4V recognizes all the illusion cases and knows their names. We opted for web data over synthetic generations to capture the rich diversity of real-world optical illusions. To mitigate the risk of data contamination, we filter the dataset using GPT4V. Specifically, we asked GPT4V to describe the image and inspect its response. We did not include an optical illusion in IllusionVQA if GPT4V could detect and describe it accurately. In some cases, GPT4V was able to detect the presence of an optical illusion but could not describe it (e.g. it mistook an Impossible Cube (Penrose & Penrose, 1958) as a Necker Cube (Rosenholtz, 2011)). We included such cases in IllusionVQA. This leaves us with 374 images of high-quality optical illusions, all of which pass the internal filter checks of the latest GPT4V and Gemini-Pro APIs. The source for each image is included in our dataset repository. We highlight a few examples of filtered-out images in Appendix I.

### 3.2 QA Generation for IllusionVQA-Comprehension

We pose the comprehension task in a standard multiple-choice VQA setting similar to (Gurari et al., 2018; Goyal et al., 2017). In some examples, this involves simply realizing that there is an illusion (Figure 2a). In other cases, we ask VLMs specific questions about the image to probe their understanding of illusions (Figure 2c).

We construct suitable questions that probe a VLM's understanding of optical illusions beyond simply detecting their presence. Each question in IllusionVQA has one correct answer and multiple wrong options. We write the questions and the correct answers and include definitively incorrect alternate options, rather than under-specified or ambiguous. We use the following methods and heuristics to generate the wrong choices:

1. We include the most likely misinterpretation of an optical illusion as an alternate whenever possible (e.g. the *"Five, the center and the four corner regions"* option in Figure 2b).
2. We prompt VLMs with the image, the question, and incorrect descriptions generated by different models as alternate options.

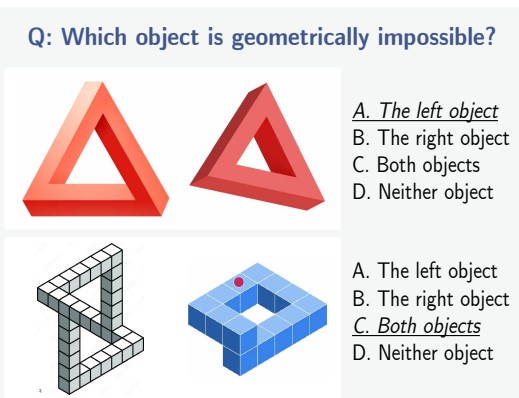

Figure 4: Examples of IllusionVQA-Soft-Localization.

3. Since VLMs are prone to ignoring visual input (Goyal et al., 2017; Jabri et al., 2016), we include incorrect descriptions generated by VLMs when prompted with only the question and without the image.
4. We manually inspect and edit each incorrect option to ensure they are challenging, given the question and the image.

We create multiple questions for an image when appropriate and ensure that the questions are distinct and not simply rephrased versions of one another. We use the Perspectives API[2] to filter out toxic content in the questions, choices, and answers. Altogether, we created 439 question-answer pairs from the 374 carefully selected optical illusion images. Through rigorous filtering and question-answer generation techniques, we ensured every instance in the IllusionVQA dataset is highly challenging yet unambiguous. Our dataset includes 30 images similar to those present in the GVIL (Zhang et al., 2023) and HallusionBench (Liu et al., 2023a) but differs in the specific question asked. Furthermore, our dataset evaluates VLMs through multiple-choice QA instead of Boolean QA in GVIL and HallusionBench.

### 3.3 IllusionVQA-Comprehension Classification

We adapt the classification methods proposed by Gregory (1997b) and Bach & Poloschek (2006) to our specific dataset and classify 12 classes as shown in Figure 3. We opt for an appearance-based classification that better fits our goal of studying VLMs in a black-box setting, without access to weights, gradients, and activations. There is no clear consensus on classifying optical illusions, so some overlap between classes was unavoidable. In particular, the Deceptive Design category could be considered a subset of the Real Scene category and the same goes for Edited Scene for the Impossible Objects category. However, the images were visually distinct enough to warrant separate categorization. Moreover, the Deceptive Design, Edited Scene, and Upside Down categories are not generally considered as optical illusions in Cognitive Science. We include these images because they are often considered optical illusions colloquially. We discuss the relevant literature on optical illusion and its classification in Appendices A and C.

### 3.4 Procedural QA Generation for IllusionVQA-Soft-Localization

We procedurally combine images of "Impossible Objects" (Figure 3) to test if VLMs can locate illusions within a scene. Although VLMs have shown impressive capability in generating bounding boxes (Bai et al., 2023; Wang et al., 2023), GPT4 and Gemini-Pro were not designed for this task. We opt for an easier localization task by stitching together two images of geometrically impossible objects horizontally and prompting the VLMs to locate the impossible object (Figure 4). We refer to this task as Soft-Localization - since the VLMs need only respond with "left" or "right" (as opposed to Precise Localization where a model outputs exact bounding box coordinates)

---

[2]https://www.perspectiveapi.com/

To evaluate the ability of VLMs on Soft Localization, we curate 40 diagrams of geometrically impossible objects and 20 diagrams of ordinary geometric objects. We further group ordinary and impossible objects that look similar as shown in Figure 4. We generate 250 images for each of the four possible answers for 1000 images in total, prioritizing similar-looking pairs. As a result, IllusionVQA-Soft-Localization consists of 1000 label-balanced samples.

Soft localization of geometrically impossible objects is challenging for VLMs because it requires precise detection of contours and advanced geometric and spatial reasoning. We conduct control experiments for soft localization and verify the ability of VLMs at this task with ordinary, yet moderately challenging objects in the Appendix F.

## 4 Experimental Setup

Since open-source VLMs do not support multiple image input, we evaluate open-source VLMs in the 0-shot and closed-source VLMs in both 0-shot and 4-shot settings. For IllusionVQA-Comprehension, we selected one sample from the four most common illusion categories as a few-shot example. The IllusionVQA-Comprehension test set contains 435 instances (370 images). Due to the inherent difficulty in describing optical illusions in natural language, we did not attempt Chain-of-Thought (CoT) evaluation on the comprehension task. For IllusionVQA-Soft-Localization, we selected one instance of each of the four alternate choices as a few-shot example. We provide a standardized reasoning template (see Appendix E) for each example in the case of 4-shot+CoT evaluation. We use the default API arguments and generation parameters.

We resize all images to 512 pixels for all models while maintaining the aspect ratio. For the soft-localization task, we apply a grayscale transform since the image color is not related to the geometry of an object. We used the latest versions of GPT4V and Gemini-Pro available at the time of writing (March 2024). We included updated (July 2024) results from GPT4o, Claude-3.5-Sonnet, and other VLMsin Appendix J. We tested three open-source VLMs, namely InstructBLIP (Dai et al., 2024), LLaVA-1.5 (Liu et al., 2023b), and CogVLM (Wang et al., 2023).

**Human Evaluation** We employed three human evaluators and provided them with all samples from the comprehension task, as well as 200 randomly chosen samples from the localization task. This was facilitated via a WebUI that tracks the time spent on each question in the background. Further details can be found in Appendix G.

## 5 Results

### 5.1 IllusionVQA-Comprehension

Table 1 shows that the performance of VLMs on illusion comprehension lags significantly behind that of humans in all categories. The larger models outperform the smaller VLMs in most categories, and GPT4V outperforms other VLMs in all but three categories.

**Human Performance** The multiple-choice setting of IllusionVQA significantly eases the difficulty in describing illusions. As a result, our three evaluators achieve $> 85\%$ accuracy which increases to 91.03% with majority voting and random tie breaks. The best-performing VLM, GPT4V with 4-shot outperforms human evaluators in only two categories, namely *Size* and *Color*. We detail inter-evaluator agreement and response times in Appendix G.

**GPT4V outperforms other VLMs.** GPT4V leads other VLMs in both 0-shot and 4-shot settings by a wide margin, even though it falls far short of human performance. We conducted further case analysis into GPT4V's output. GPT4V leads in ten of the twelve IllusionVQA categories and shows competitive performance in the remaining two. Categories such as Real-Scene, Size, Color, and Deceptive Design test a VLM's perception of depth, color, and challenging scenes; skills that are particularly relevant when using VLMs as autonomous robots in the real world (Zitkovich et al., 2023; Wake et al., 2023). GPT4V maintains sub-

| Class | # | 0-shot | | | | | 4-shot | | Human |
|-------|---|--------|--|--|--|--|--------|--|-------|
| | | I-BLIP | LLaVA | Cog | Gemini | GPT4V | Gemini | GPT4V | |
| Impossible Object | 134 | 34.22 | 43.28 | 44.03 | **56.72** | 55.22 | 56.72 | 58.96 | 98.51 |
| Real-Scene | 64 | 26.56 | 42.19 | 34.38 | 46.88 | **57.81** | 46.88 | 54.69 | 98.44 |
| Size | 46 | 26.09 | 19.57 | 13.04 | 45.65 | **58.70** | 52.17 | 69.57 | 63.04 |
| Hidden | 45 | 44.44 | 42.22 | 42.22 | 42.22 | **51.11** | 48.89 | 46.67 | 100 |
| Deceptive Design | 37 | 37.84 | 43.24 | 45.95 | 64.86 | **70.27** | 67.56 | 72.97 | 94.59 |
| Angle Illusion | 26 | 30.77 | 38.46 | 30.77 | 53.85 | **69.23** | 50 | 84.62 | 84.62 |
| Color | 23 | 30.43 | 26.09 | 30.43 | 17.39 | **69.57** | 17.39 | 82.61 | 60.87 |
| Edited-Scene | 21 | 42.86 | 61.90 | 42.86 | 66.67 | **71.43** | 66.67 | 80.95 | 100 |
| Upside-Down | 7 | 42.86 | **71.43** | **71.43** | 57.14 | **71.43** | 57.14 | 71.43 | 100 |
| Pos.-Neg. Space | 7 | 57.41 | 42.86 | 71.43 | **85.71** | 57.14 | 71.43 | 85.71 | 100 |
| Circle-Spiral | 6 | 33.33 | 0.00 | 16.67 | 33.33 | **50** | 33.33 | 33.33 | 66.67 |
| Miscellaneous | 19 | 36.84 | 42.11 | 42.11 | **52.63** | 42.11 | 57.89 | 42.11 | 89.47 |
| Total | 435 | 34.25 | 40 | 38.16 | 51.26 | **58.85** | 52.87 | 62.99 | 91.03 |

Table 1: Performance of VLMs on IllusionVQA-Comprehension. Categories where accuracy has improved using 4-shot prompting are underlined. 'Human' performance refers to the aggregated performance based on the majority vote of three human evaluators.

stantial leads in all four categories and is fairly resistant to challenging real scenes, with a 10.93% lead over Gemini-Pro.

**Small VLMs**   Among the three small VLMs (#parameters $\sim 14 - 17B$), LLaVA-1.5 and CogVLM show similar performance in most categories while InstructBLIP shows the largest variation in different categories, performing significantly better (>5%) in comprehending *Size* and *Circle-Spiral* illusions while performance significantly worse in other categories.

**In-Context Learning (ICL)**   We tested GPT4V and Gemini-Pro using 4-shot learning and observed marginal improvements in overall accuracy in IllusionVQA-Comprehension. However, the improvement was not consistent between different categories. For example, 4-shot GPT4V accuracy was lower than 0-shot in three categories, namely Real-Scene, Hidden Illusion, and Circle-Spiral Illusions. Gemini-Pro shows similar accuracy drops but in different categories. It is possible that ICL is not a ubiquitous strategy because few-shot examples introduce additional language priors that bias the VLMs towards incorrect answers (Goyal et al., 2017; Jabri et al., 2016).

**Performance of VLMs on Ordinary Vertically-Flipped Images**   The *Upside-Down* category for images that depict two or more intertwined entities depending on vertical orientation. As shown in Table 1, the smaller open-source VLMs perform competitively in this category while significantly lagging behind closed-source VLMs in most others. To validate this discrepancy, we curated a subset of VQA-v2.0 (Goyal et al., 2017) and tested LLaVA-1.5 and Gemini-Pro on both unedited and vertically flipped images. Table 2 shows that Gemini-Pro's accuracy drops by 15.5% on flipped images, while LLaVA-1.5's accuracy only drops by 6.5%, confirming our findings in Table 1.

| VLM | Orientation | Accuracy |
|-----|-------------|----------|
| LLaVA-1.5 | Normal | 80 |
| Gemini-Pro | Normal | 79.5 |
| LLaVA-1.5 | Vertically Flipped | 73.5 |
| Gemini-Pro | Vertically Flipped | 64 |

Table 2: Performance of VLMs on a subset of 200 instances from VQA-v2.0 (Goyal et al., 2017). The model outputs were manually evaluated.

## 5.2   IllusionVQA-Soft-Localization

Table 3 shows that the localization of optical illusions is a significant challenge for VLMs but quite straightforward for humans. Small open-source VLMs, particularly, show close

| VLM | Prompt Type | Accuracy |
|---|---|---|
| InstructBLIP | 0-shot | 24.3 |
| LLaVA-1.5 | 0-shot | 24.8 |
| CogVLM | 0-shot | 28 |
| GPT4V | 0-shot | 40 |
| | 4-shot | 46 |
| | 4-shot + CoT | 49.7 |
| Gemini Pro | 0-shot | 43.5 |
| | 4-shot | 41.8 |
| | 4-shot + CoT | 33.9 |
| Human | | 100 |

Table 3: Performance of VLMs on IllusionVQA-Soft-Localization. Human performance denotes the majority-vote performance of three human evaluators on 200 random instances.

to random performance (25%). GPTV and Gemini-Pro fare somewhat better but fall short of human performance. On 200 randomly selected examples, all three human evaluators achieved perfect accuracy. Even with ICL and CoT, the performance of GPT4V in the IllusionVQA-Soft-Localization dataset remains subpar, achieving only 49.7% accuracy[3]. This unexpectedly poor performance motivated us to conduct additional experiments, presented in Appendix F.

**Large VLMs Can Localize Ordinary Objects But Not Illusions** We created a new dataset for soft localization using different brands of sports cars and asked the VLM "Which car is a [BRAND]?" with the same four options as in Figure 4). This task remains moderately challenging for VLM since sports cars of different brands share the same basic shape and differ in details. Although smaller VLMs struggled with this task, both GPT4V and Gemini-Pro achieved 100% accuracy. GPT4V and Gemini's stellar performance in soft localization of ordinary objects makes their unexpectedly low performance on IllusionVQA-Soft Localization surprising. This is likely because differentiating geometric illusions from ordinary objects requires spatial reasoning (a known weakness of VLMs (OpenAI, 2023)) instead of feature extraction. We present the details regarding the dataset and the model performance in Appendix F.

**In-Context Learning and Chain-of-Thought** In-context learning (ICL) (Brown et al., 2020) and Chain-of-Thought (CoT) (Wei et al., 2022) are widely used techniques that boost LLM performance. Research has shown that these techniques are effective in vision language tasks (Yang et al., 2023), but are still understudied.

Table 3 shows that Gemini-Pro performs better than GPT4V (+3.5%) when evaluated without ICL or CoT. However, GPT4V shows the highest accuracy overall when evaluated with 4-shot+CoT (+9.7% gain) while Gemini-Pro's performance degrades substantially (-9.6%). We hypothesize this is because GPT4V is likely the larger model and exhibits stronger emergent ICL abilities (Wei et al., 2023). As illustrated in Figure 5, there are a substantial number of instances in which the 0-shot evaluation yielded the correct answer, while 4-shot+CoT reasoning failed to do so. This implies that ICL and CoT are not universally applicable to IllusionVQA-Soft-Localization.

**Failure on In-Context Examples** To probe the weakness of VLMs, we conducted additional experiments in which we asked GPT4V and Gemini-Pro with 4 examples (4 shots) and presented the same examples as questions. In this case, the exact image, the question, and the correct answer are in the context window of the VLM. Surprisingly, neither GPT4V nor Gemini-Pro can successfully answer all questions. This highlights a puzzling limitation of ICL for VLMs where models fail even when the correct answer is in the context which points to over-dependence on language priors (Goyal et al., 2017; Jabri et al., 2016) and visual information is all but ignored. This phenomenon persists in all the combinations we tried. We provide the prompt and result for one such combination in Appendix H.

---

[3]Random guessing would yield a 25% accuracy rate.

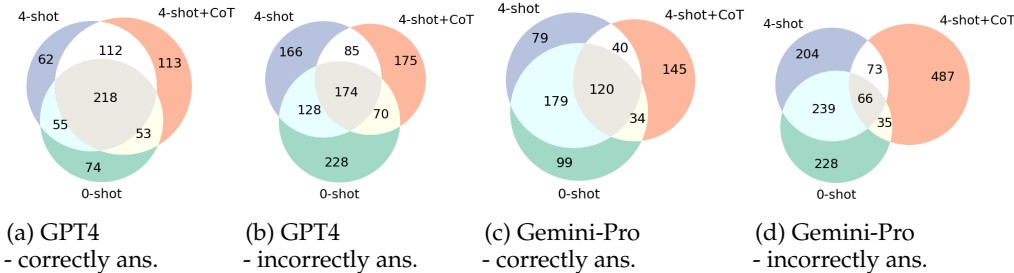

Figure 5: Venn Diagrams showing the agreement between prompting techniques. There are instances where ICL and CoT cause the VLMs to answer incorrectly.

## 6 Discussion

### 6.1 Illusion-Resistant VLMs for Robotics

Vision Language Models (VLMs) are increasingly being deployed in real-world applications, particularly in robotics (Zitkovich et al., 2023; Wake et al., 2023), where they play a crucial role in enabling robots to navigate and interact with their environment. Embodied AI need to demonstrate consistent performance across a wide range of real-world situations, including those involving deceptive designs, optical illusions and obscured objects.

The IllusionVQA dataset has the potential to function as a complete stress test to assess the robustness and adaptability of embodied VLMs. By assessing the performance of VLMs on this challenging dataset, researchers can pinpoint areas of weakness and enhance the models' capability to navigate perceptually ambiguous scenarios.

### 6.2 Thinking Fast and Slow - Comparing VLM and Human Response Times

Kahneman (2011) popularized the concept of two modes of thinking: 'System 1', which is fast and instinctive, and 'System 2', which is slower, more deliberate, and logical. Due to their autoregressive architecture, current state-of-the-art LLMs are primarily capable of 'System 1' thinking (Bubeck et al., 2023). Approximating 'System 2' reasoning with LLMs has motivated significant research efforts such as Chain-of-Thought (Wei et al., 2022), Tree-of-Thought (Yao et al., 2024), and other related methods.

Understanding and locating optical illusions requires deliberate and logical 'System 2' thought processes. Human evaluators spent an average of 14.99 seconds on each question from IllusionVQA-Comprehension and 5.68 seconds on IllusionVQA-Soft-Localization. Since GPT4V and Gemini-Pro APIs don't reveal exact inference times, we report API response times[4]. In the 0-shot (4-shot) setting, GPT4V takes an average of 1.81 (3.27) seconds, and Gemini-Pro takes 3.82 (4.59) seconds regardless of the task. While API response times significantly overestimate model inference times, they are still notably faster than the time a human spends deliberating on each question.

## 7 Conclusion

We demonstrate that current state-of-the-art VLMs struggle with understanding optical illusions and largely falter in identifying geometrically impossible objects. We find that GPT4V maintains substantial leads in most illusion types and that there is a noticeable performance gap between open-source and closed-source VLMs. We validate the effectiveness of few-shot learning for GPT4V and Gemini-Pro in the comprehension task but show that few-shot learning and Chain-of-Thought reasoning are not universally effective in the localization task. Additionally, we discover that GPT4V and Gemini-Pro fail to correctly answer illusion localization questions even when the correct answer is provided in the context. We release our dataset and evaluation code to stimulate further research.

---

[4]This includes image-upload times and internal load-balancing delays, likely making API response times much slower than internal model inference times.

## Limitation

Our current study presents valuable initial findings; however, we acknowledge several limitations that present avenues for future research.

1. **Scale of the dataset:** The number of examples in the IllusionVQA-Comprehension dataset is relatively modest due to the rigorous filtering process applied to the initial candidate set of over 3500 images scraped from the Internet. Despite extensive efforts, we could not find additional optical illusions that met our inclusion standards. One potential avenue for expanding the dataset in the future could be the use of synthetic optical illusions. However, the current capabilities of image generation models in creating novel, high-quality optical illusions are limited, making this approach challenging for the time being.
2. **Evaluation Scope:** Due to the OpenAI API rate limits (500 requests/day), we were unable to evaluate GPT4V in multiple evaluation runs. It is possible that more advanced evaluation techniques (Lei et al., 2024) or task-specific fine-tuning of open-source VLMs with sufficient data might yield improvements on IllusionVQA-Soft-Localization.
3. **Open-ended Question Answering:** We primarily refrained from investigating open-ended QA due to the subpar performance of our baseline multiple-choice VQA setup. The complexity of open-ended generation is further heightened by the need for additional human evaluators to serve as judges.

## Acknowledgement

We thank the contributors and the owner of moillusions.com for curating and hosting a collection of high-quality optical illusions online. We thank Michael Bach for allowing us to gather optical illusions and their descriptions from his excellent repository. Lastly, we appreciate the efforts of our human evaluators who diligently evaluated the IllusionVQA dataset.

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

# Supplementary Material: Appendices

## A  Preliminaries

This section covers the relevant cognitive science literature regarding optical illusions.

**Optical Illusion**    An optical illusion refers to a visual perception phenomenon in which the visual system misinterprets reality, resulting in a disparity between what is perceived and what exists. Researchers have proposed that optical illusions mainly arise from discrepancies in a person's perceptual knowledge and conceptual knowledge (Gregory, 1997b; Carbon, 2014). As a concrete example, our fast perceptual knowledge perceives Figure 2a as an elephant, yet upon closer inspection, we understand that the diagram does not align with our preexisting concepts regarding geometry.

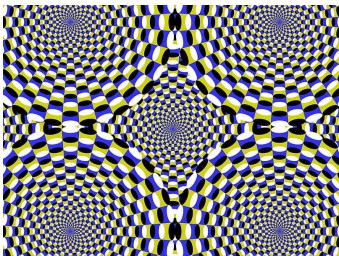 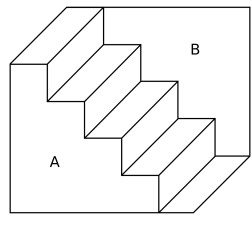

Figure 6: Left: Kitaoka's 'Throwing cast nets' (Conway et al., 2005), an example of cognitive impenetrability. Even though we know it is a static image, the illusion of motion persists. Right: Schroeder stairs (Rosenholtz, 2011), an example of an ambiguous or bistable image. The drawing may be perceived as either a staircase leading from left to right downward or as the same staircase only turned upside down.

**Cognitive impenetrability**    is a property of the basic information processes of cognition. If a process is cognitively impenetrable, then the operation of this process cannot be altered by changing the contents of mental representations (i.e., an agent's beliefs, desires, or goals). This is because the process is *"wired in"* and its operation must be explained by appealing to neuroscience (Dawson, 2017). Select visual illusions (Figure 6 - Left) exhibit this phenomenon because the illusion persists even if we know or are made aware of the illusion.

**Ambiguous or Bi-stable Images**    are images that create ambiguity between two or more distinct image forms. Classic examples of ambiguous images include Rubin's Vase (Hasson et al., 2001), Schroeder stairs (Figure 6 - Right), and Necker's Cube (Rosenholtz, 2011) among many others.

## B  Object Localization with VLMs

Object Detection (Girshick et al., 2014) is a core computer vision task in which models output the bounding box coordinates of objects in an image. Although pure vision models such as DETR(Zong et al., 2023) are state of the art in object detection, there has been considerable progress in object detection using pretrained VLMs (Wang et al., 2023; Bai et al., 2023). However, most VLMs do not natively support object detection and perform significantly worse than vision-only models. Consequently, we opt for a simpler task, soft localization, where the VLMs are prompted to output the approximate position of objects (i.e. left side or right side) instead of exact bounding box coordinates.

## C    Literature Review of Illusion Classification

Classifying optical illusions has posed significant challenges, with no definitive consensus reached within the scientific community. We review two prominent categorization approaches proposed by Gregory (1997b) and Bach & Poloschek (2006) as precursors to our categorization of IllusionVQA.

Gregory (1997b) proposed a four-fold categorization of optical illusions based on their underlying causes: physical, physiological, knowledge-based, and rule-based. Physical illusions, like those induced by fog or lens distortions, directly interfere with image formation. Physiological illusions, such as scintillations from pressure on the eye, arise from disruptions within the visual system. Knowledge-based illusions, exemplified by interpreting 2D sketches as 3D objects, leverage prior knowledge to misinterpret sensory input. Finally, rule-based illusions, such as Kanizsa triangles (Kanizsa, 1976), exploit the visual system's tendency to complete patterns or infer missing information, leading to illusory percepts. Gregory (1997b) proposes a complementary classification of optical illusions based on appearance and rooted in natural language, namely ambiguities (Figure 6), distortions (Figure 2b), paradoxes (Figure 2a), and fictions (Figure 2c). We adopt this classification scheme for IllusionVQA in section 3.3.

## D IllusionVQA-Comprehension Categories and Question Types

| Class | # | Description | Example |
|-------|---|-------------|---------|
| Impossible Object | 134 | Objects that cannot exist in 3D space. | Penrose Triangle (Penrose & Penrose, 1958) |
| Real-Scene | 64 | Hard to interpret due to forced perspective, unnatural poses, objects being obscured, etc. | |
| Size | 46 | Two objects with the same dimensions look different or vice versa. | Müller-Lyer illusion (Judd, 1905) |
| Hidden | 45 | (Silhouettes of) certain objects are present somewhere without being immediately obvious. | |
| Deceptive Design | 37 | Items have been painted to appear as something else. | Trompe-l'œil art (Wade & Hughes, 1999) |
| Angle Illusion | 26 | Parallel lines look curved or angled, and vice versa. | Zöllner illusion (Zöllner, 1860). |
| Color | 23 | Different colors or shades appear the same, and vice versa. | Checker Shadow Illusion (Adelson, 1995) |
| Edited-Scene | 21 | Edited to depict something impossible although it may appear normal on a cursory view. | |
| Upside-Down | 7 | Two intertwined entities depending on vertical orientation. | |
| Positive-Negative Space | 7 | Images that create ambiguity between two or more distinct forms represented by two or more colors. | Rubin's vase (Hasson et al., 2001) |
| Circle-Spiral | 6 | Circles appear as spirals due to peculiar color or background pattern. | Fraser Spiral (Fraser, 1908) |
| Miscellaneous | 19 | | |
| Total | 435 | | |

Table 4: Types of Illusions in IllusionVQA-Comprehension - Test Split

| Major Question Types | Count |
|----------------------|-------|
| *"What is unusual about this image?"* | 114 |
| *"Describe this image."* | 80 |
| *"What is hidden in this image?"* | 22 |
| Image-specific Questions | 219 |

Table 5: Question Types in IllusionVQA-Comprehension - Test Split.

## E  Few-shot and Chain-of-Thought Evaluation Prompts

**0-shot Instruction**  -

```
You'll be given an image, an instruction, and some options. You have to select
    the correct one. Do not explain your reasoning. Answer with only the letter
     that corresponds to the correct option. Do not repeat the entire answer.
```

**4-shot Instruction**  -

```
You'll be given an image, an instruction, and some options. You have to select
    the correct one. Do not explain your reasoning. Answer with only the letter
     that corresponds to the correct option. Do not repeat the entire answer.
    Here are a few examples: {few-shot examples} Now you try it.
```

**4-shot+CoT Instruction**  -

```
You'll be given an image, an instruction, and some choices. You have to select
    the correct one. Reason about the choices in the context of the question
    and the image. End your answer with 'Answer': {letter_of_correct_choice}
    without the curly brackets. Here are a few examples: {few-shot examples}
    Now you try it.
```

**CoT Reasoning Template**  -

```
The left object is a {description} which is {possible/impossible}. The right
    object is a {description} which is {possible/impossible}."}
```

## F  Soft Localization with Ordinary Objects

Since the performance of VLMs on the Soft Localization task of IllusionVQA had been unexpectedly poor, we created a simple auxiliary dataset to test the capability of VLM in locating ordinary objects. Specifically, we use the images of different brands of sports cars and ask the model to locate a certain brand. Since the two objects are visually similar, the task remains moderately challenging.

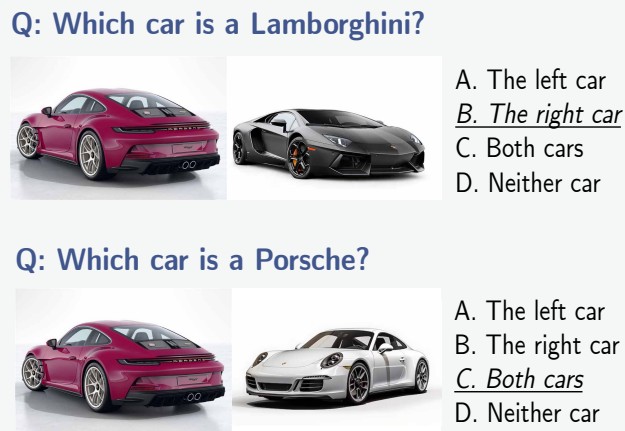

Figure 7: Soft localization of ordinary objects.

Figure 7 shows samples from this new dataset. We collect 400 instances for this task and test all VLMs in the 0-shot setting. Table 6 shows that while small VLMs perform poorly on this task, GPT4V and Gemini-Pro achieve perfect accuracy. This confirms that large VLMs can locate objects in a scene.

| VLM | Prompt Type | Accuracy |
|---|---|---|
| InstructBLIP | 0-shot | 30 |
| LLaVA-1.5 | 0-shot | 29.25 |
| CogVLM | 0-shot | 49 |
| Gemini-Pro | 0-shot | 100 |
| GPT4V | 0-shot | 100 |

Table 6: Accuracy of localizing ordinary objects.

## G  Human Evaluation

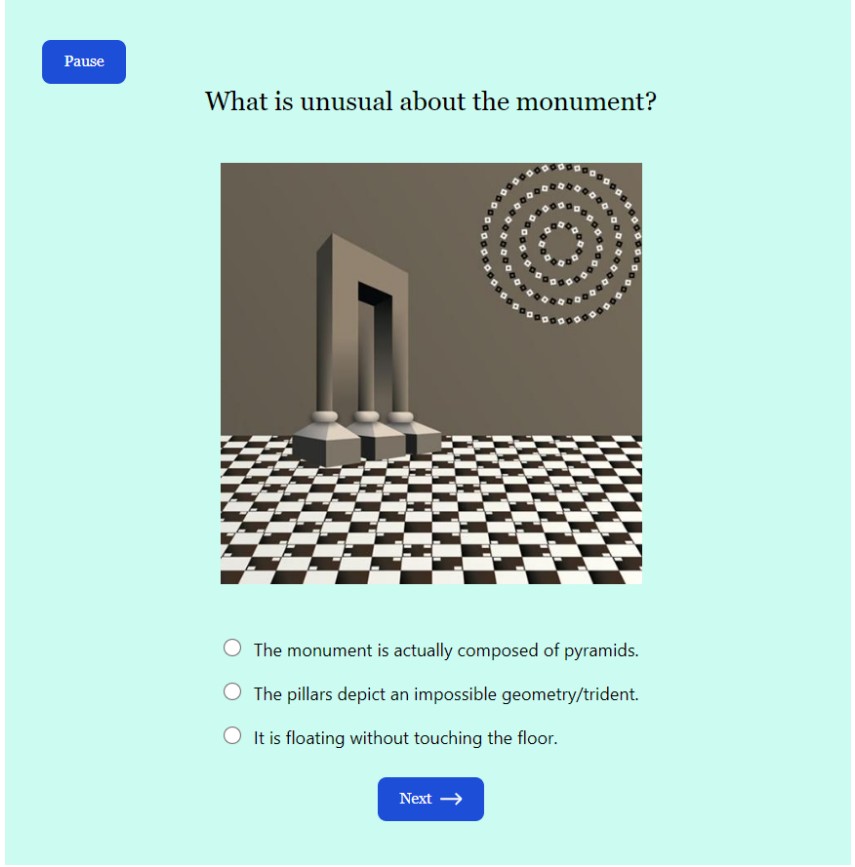

Figure 8: WebUI presented to human evaluators.

We employed three nonauthor undergraduate students from the author's institution to evaluate how well humans do on comprehension and localization of optical illusions. The evaluation was carried out without Internet access and the evaluators were not informed in advance that the task was related to optical illusions. We presented each evaluator with a WebUI (Figure 8) and with the following instructions.

**Instructions to Evaluators:** *'Each page will have a question and an image of an optical illusion followed by some options. Select the option that best answers the question. Illusions can be misleading, so choose the option that reflects how entities in the image truly are, not how they might appear. Zoom in or out to get a better view, if necessary. Each response is being timed internally, so use the "Pause" button when required.'*

All three evaluators were presented with the questions in the same order. Table 7 shows that the individual accuracy of the evaluators is comparable. The pairwise Cohen's Kappa

| Evaluator | Accuracy | Avg. Response Time |
|:---:|:---:|:---:|
| 1 | 87.13 | 16.46 |
| 2 | 86.67 | 12.95 |
| 3 | 92.87 | 15.57 |

Table 7: Human performance and response time on IllusionVQA-Comprehension.

between the evaluators are 0.808 0.796 and 0.773, indicating a substantial level of inter-evaluator agreement [5]. On IllusionVQA-Soft-Localization, all evaluators achieved perfect accuracy on the selected subset of 200 instances and the average response time was 5.68 seconds.

## H    Limits of In-Context Learning For Optical Illusions

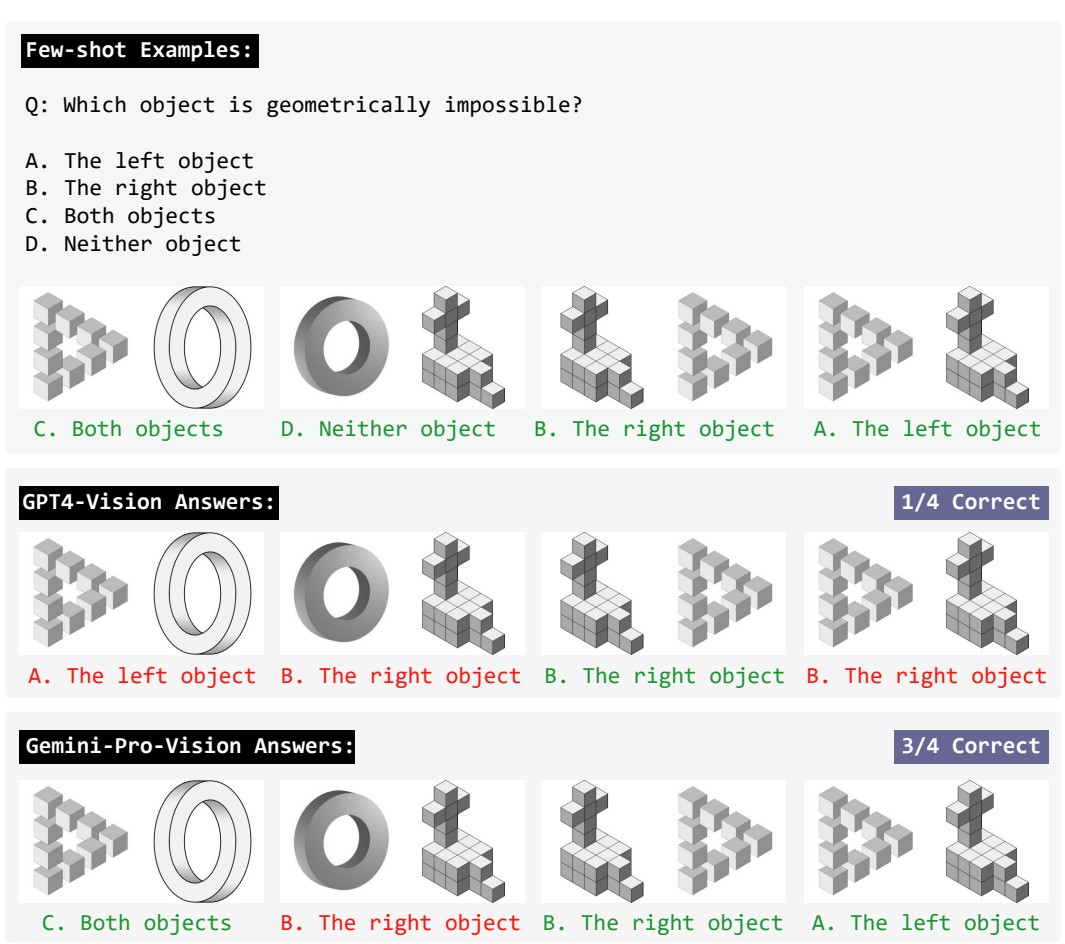

Figure 9: GPT4V and Gemini-Pro consistently fail to locate geometrically impossible objects even when the correct answer is in context (4-shot).

---

[5]To determine the agreement metric, we considered each question-option pair as a singular question with a yes/no answer which denotes whether an evaluator has selected that option as the answer or not.

## I  Examples of Images Removed During Preprocessing

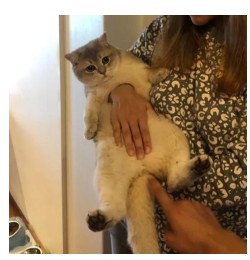 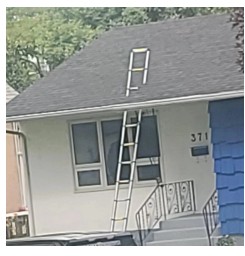 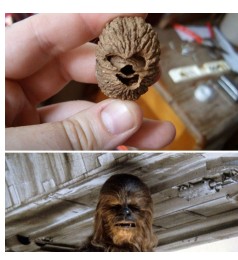 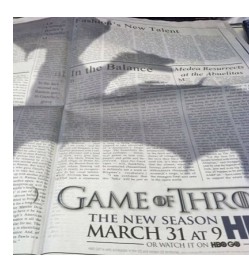

(a) Unclear if edited     (b) Unclear if edited     (c) Pareidolia     (d) Printed shadow design

Figure 10: Examples of images we filtered out during pre-processing. These images were found in online archives of optical illusions and are colloquially considered illusions.

## J  Updated IllusionVQA-Comprehension Results

| Class | # | 0-shot | | | | | | 4-shot | | | | Human |
|---|---|---|---|---|---|---|---|---|---|---|---|---|
| | | InternVL2 8B | Pali Gemma | Gemini 1.0-Pro | Claude 3.5 Sonnet | GPT4V | GPT4o | Gemini 1.0-Pro | Claude 3.5 Sonnet | GPT4V | GPT4o | |
| Impossible Object | 134 | 49.25 | 32.09 | 56.72 | **64.93** | 55.22 | 63.43 | 56.72 | 63.43 | 58.96 | 61.94 | 98.51 |
| Real-Scene | 64 | 40.63 | 35.94 | 46.88 | 54.69 | 57.81 | **64.06** | 46.88 | 57.81 | 54.69 | 57.81 | 98.44 |
| Size | 46 | 43.48 | 15.22 | 45.65 | 50.00 | **58.70** | 45.65 | 52.17 | 80.43 | 69.57 | 93.47 | 63.04 |
| Hidden | 45 | 44.44 | 33.33 | 42.22 | 37.78 | 51.11 | **66.67** | 48.89 | 44.44 | 46.67 | 48.89 | 100 |
| Deceptive Design | 37 | 37.84 | 32.43 | 64.86 | 70.27 | 70.27 | **72.97** | 67.57 | 67.57 | 72.97 | 78.38 | 94.59 |
| Angle Illusion | 26 | 50.00 | 26.92 | 53.85 | **73.08** | 69.23 | 50.00 | 50.00 | 73.08 | 84.62 | 80.77 | 84.62 |
| Color | 23 | 26.09 | 34.78 | 17.39 | 65.22 | **69.57** | 52.17 | 17.39 | 86.96 | 82.61 | 78.26 | 60.87 |
| Edited-Scene | 21 | 66.67 | 42.86 | 66.67 | 61.90 | 71.43 | **80.95** | 66.67 | 71.43 | 80.95 | 85.71 | 100 |
| Upside-Down | 7 | **85.71** | 42.86 | 57.14 | **85.71** | 71.43 | 71.43 | 57.14 | 85.71 | 71.43 | 42.86 | 100 |
| Pos.-Neg. Space | 7 | 42.86 | 42.86 | **85.71** | 71.43 | 57.14 | **85.71** | 71.43 | 71.43 | 85.71 | 71.43 | 100 |
| Circle-Spiral | 6 | 0.00 | 0.00 | 33.33 | 33.33 | **50.00** | **50.00** | 33.33 | 50.00 | 33.33 | 50.00 | 66.67 |
| Miscellaneous | 19 | 42.11 | 31.58 | **52.63** | 47.37 | 42.11 | **52.63** | 57.89 | 52.63 | 42.11 | 52.63 | 89.47 |
| Total | 435 | 45.06 | 31.26 | 51.26 | 59.08 | 58.85 | **62.53** | 52.87 | 66.44 | 62.99 | 67.12 | 91.03 |

Table 8: IllusionVQA-Comprehension results updated to include state-of-the-art VLMs as of July 2024. Added evaluation of InternVLM2 (Chen et al., 2024), PaliGemma-3B (Beyer et al., 2024), GPT-4o (OpenAI, 2024) and Claude 3.5 Sonnet Anthropic (2024). Categories where accuracy has improved using 4-shot prompting are underlined.

