# OpenReview forum: "IllusionVQA: A Challenging Optical Illusion Dataset for Vision Language Models"
_colmweb.org/COLM/2024/Conference — COLM_

### Official Review · Reviewer_7WKq · 2024-04-26

**Rating:** 6
**Confidence:** 4
**Ethics Flag:** 1

**Summary:**

This paper introduce IllusionVQA, a novel dataset designed to rigorously test the ability of VLMs to locate and comprehend challenging optical illusions. The authors comprehensively test a wide range of open-source and closed-source VLMs and find some interesting results.

**Questions To Authors:**

Please list more difference with Hallusionbench.

**Reasons To Accept:**

The dataset is interesting and challenging enough. The results are good.

**Reasons To Reject:**

The paper is very similar to hallusionbench. The only difference is that Hallusionbench is binary answer while IllusionVQA is multiple choice questions.

---

> ### Author Rebuttal · Authors · 2024-05-30
>
> **Dear Reviewer  7WKq**\
> Thank you for your review. We appreciate the opportunity to respectfully highlight the contributions of our work and the differences from other datasets, and we hope this addresses your concerns. We would kindly invite further discussion, as we believe our dataset is quite different from existing benchmarks.
>
> **Concern [1]: *“The only difference is that Hallusionbench is binary answer while IllusionVQA is multiple choice questions.”***\
> While HallusionBench is an excellent investigation of VLM capabilities, there are major differences in the dataset, methodology, and overarching goal of our work.
>
> **1. Investigation Goal:** HallusionBench aims to answer *“when and why VLMs hallucinate”* while our work investigates *“how well VLMs can deduce objective reality from images of optical illusions”*. The authors of HallusionBench did not differentiate between optical illusions and other image types such as text and posters during evaluation.
>
> **2. Dataset Size and Diversity**: IllusionVQA-Comprehension (374 images and 12 categories) is much larger and more diverse than HallusionBench-Illusions (72 root images and 2 categories).
>
> **3. Data Contamination:** The authors of HallusionBench note that GPT-4V recognized the optical illusions in all 72 root images in their dataset. We filtered out any image that GPT-4V was able to describe without options, thus largely mitigating dataset contamination.
>
> **4. Human Evaluation:** HallusionBench aims to test VLM hallucination, so they do not perform a controlled human evaluation on their dataset. In contrast, we employed three human evaluators to establish that VLMs are still far behind humans when deducing objective reality in most illusions.
>
> **5. Soft Localization:** Soft-localization of geometrically impossible objects is a significant and novel part of our paper’s contribution. Determining which object is geometrically impossible is a complex task that requires a deep understanding of 3D geometry and isometric perspectives.  In Table 3, we show that all VLMs struggle with soft-localization while humans achieve perfect accuracy.
>
> **6. Advanced Prompting Techniques:** To the best of our knowledge, we are the first to test the effect of In-Context Learning (ICL) and Chain-of-Thought (CoT) prompting on optical illusion tasks for VLMs. We discover that ICL and CoT are not universally effective and degrade the performance of Gemini-Pro on IllusionVQA-Soft-Localization  (Please see Table 3).

---

> ### Author Response · Authors · 2024-06-06
> **Follow up with Reviewer 7WKq**
>
> Dear Reviewer 7WKq,
>
> We would love to hear back from you and make sure that all your concerns are addressed.
>
> Sincerely,
>
> Authors

---

### Official Review · Reviewer_L4KJ · 2024-05-06

**Rating:** 5
**Confidence:** 4
**Ethics Flag:** 1

**Summary:**

This paper introduces a new benchmark for evaluating the performance of vision-language models under visual illusions through standard multiple-choice VQA tasks. The authors measure a wide range of model performances under different settings, such as the inclusion of In-Context examples and Chain-of-Thought prompting.

**Reasons To Accept:**

1. The mined dataset of 374 examples can be a useful resource for the community.
2. The study on the effects of in-context examples and chain-of-thought prompting reveals interesting phenomena, such as in-context examples not always being helpful.

**Reasons To Reject:**

1. Data contamination and bias in the dataset: GPT-4V is used as a filter in the dataset creation process, which likely results in a dataset biased towards GPT-4V's capabilities. Additionally, the majority of the data is sourced from the web, which likely comes with data contamination issues and is already part of many models' pre-training data. In comparison, prior works like GVIL/Hallusionbench use synthetic data to mitigate this problem.
2. The authors assume that the model should not experience illusions, which is still a topic open for debate. As discussed in GVIL, it remains an open question whether we want models to "experience the same kinds of illusions as humans do, or to faithfully represent reality?" This paper directly uses "the most likely misinterpretation of an optical illusion" (the one humans perceive?) as a negative example, presupposing a specific stance on this issue.

---

> ### Author Rebuttal · Authors · 2024-05-30
>
> **Dear Reviewer L4KJ**\
> Thank you for your constructive review. We appreciate the opportunity to clarify our decisions regarding dataset curation and annotation methodology.
>
> **Concern 1: Data Contamination and GPT4V bias.**\
> We fully acknowledge the risks of data contamination and bias when using web-sourced data and have taken steps to mitigate these concerns during the dataset curation process. We opted for web data over synthetic generations to capture the rich diversity of real-world optical illusions. Certain types of illusions, such as impossible objects, challenging real-world scenes, or deceptive designs, cannot be generated synthetically. IllusionVQA-Comprehension covers 12 distinct categories, making it larger and more comprehensive than previous datasets focused solely on color and size illusions.
>
> To mitigate contamination, we removed any optical illusions that the state-of-the-art GPT-4V model could answer correctly zero-shot without being provided options.  This filtration step ideally removes illusions for which GPT-4V memorized the correct explanations. Since GPT-4V was the best-performing VLM  in our pilot experiments, its zero-shot capability was the best proxy for measuring potential data contamination. Without such model-based filtration, there would be no reliable way to assess and mitigate contamination risks. Notably, this implicitly biases the dataset *against* GPT4V since only illusions it could not comprehend were included in the dataset.
>
> **Concern [2.1]: *“the most likely misinterpretation of an optical illusion" (the one humans perceive?)”***\
> Yes! We refer to human misperception in that case. We also include zero-shot wrong answers generated by VLMs as alternate options as outlined in Section 3.2.
>
> **Concern [2.2]: *“The authors assume that the model should not experience illusions…”***\
> That’s an excellent observation! We fully agree that there is no right answer for how VLMs should interpret optical illusions. As such, following HallusionBench [1],  we used "objective reality" as the benchmark for IllusionVQA. We did not include any images where we could not deduce objective reality solely from the images (Please refer to Appendix I for concrete examples). We likewise instructed human evaluators to answer according to objective reality (Please refer to Appendix G).
>
>
> [1] Liu et al. HallusionBench: An Advanced Diagnostic Suite for Entangled Language Hallucination & Visual Illusion in Large Vision-Language Models [CVPR 2024]

---

> > ### Comment · Reviewer_L4KJ · 2024-06-06
> >
> > Thank you for the response! However, I still think GPT4V bias is a concerning issue for a benchmark-centric paper and am therefore keeping my scores.

---

### Official Review · Reviewer_9wgj · 2024-05-10

**Rating:** 6
**Confidence:** 4
**Ethics Flag:** 1

**Summary:**

This paper presents a novel dataset aimed at evaluating Vision Language Models (VLMs) on their ability to understand and localize optical illusions, featuring two principal tasks: comprehension and soft localization. These tasks test VLMs' proficiency in interpreting complex visual phenomena that are typically misleading to human vision. The dataset has been rigorously tested on various state-of-the-art VLMs, including GPT4V and Gemini-Pro. While these models generally perform well in standard object recognition, they exhibit significant difficulties with optical illusions, especially in tasks demanding intricate visual and spatial reasoning.
Overall, this paper is well-structured and clear, with methodical descriptions of experimental methods, dataset construction, and results, enhancing its accessibility and understanding. However, this paper lacks detailed descriptions of data sources, raising concerns about its diversity and applicability for practical VLM applications.

**Reasons To Accept:**

The IllusionVQA dataset introduces a unique challenge to the field of VLMs by focusing on optical illusions. This dataset differs from traditional VQA datasets by specifically testing the models' comprehension and spatial reasoning abilities through its two main tasks: comprehension and soft localization. These tasks assess VLMs' capability to interpret and localize elements within images that are inherently deceptive, extending beyond basic object recognition to probe deeper into advanced visual processing.
Initial testing on the IllusionVQA dataset highlights significant shortcomings of contemporary VLMs, including GPT-4V. The underperformance compared to near-perfect human results underscores the current gap in VLMs' ability to handle complex visual inputs.These findings point to a pressing need for advancements in VLM architectures and training methods.

**Reasons To Reject:**

This paper lacks clear descriptions of the data sources for its optical illusion images, raising concerns about the diversity and representativeness of the dataset. There is no detailed methodology on ensuring data comprehensiveness, which is crucial for a varied visual category like optical illusions. Furthermore, the study does not discuss whether the illusions are representative of real-world scenarios, which would be vital for practical applications of VLMs. Addressing these issues could enhance the dataset’s utility and relevance, providing a stronger benchmark for VLMs in practical environments.

---

> ### Author Rebuttal · Authors · 2024-05-30
>
> **Dear Reviewer 9wgj**\
> Thank you for your valuable feedback. We appreciate the opportunity to clarify your concerns and improve our work based on your suggestions.
>
> **Concern 1: *“This paper lacks clear descriptions of the data sources for its optical illusion images, raising concerns about the diversity and representativeness of the dataset.”***\
> The majority of our dataset is from two online sources. 216 out of 374 images in IllusionVQA-Comprehension are from an image-sharing website about optical illusions. Most of the “colloquial” illusions in IllusionVQA ( real-scene, edited-scene, hidden, deceptive-design) are from this source. A further 27 images are from a second website maintained by an optical illusions expert. This website was the primary source for formal optical illusions (color, angle, and size categories). The remaining 131 images are from secondary sources such as Pinterest and Imgur.
>
> We have **refrained from specifically mentioning the two primary website names because we contacted the website administrators** and obtained permission to collect images. We will acknowledge their invaluable contributions, clarify data sources, and include all URLs in the final versions of our manuscript and dataset.
>
> **Concern 2: *“There is no detailed methodology on ensuring data comprehensiveness, which is crucial for a varied visual category like optical illusions.”***\
> We scraped all relevant online repositories and collected more than 3500 images, which we filtered down to 374. We encountered repetitions or trivial variations when we attempted to collect more images. Likewise, procedurally generated optical illusions are not diverse enough to warrant inclusion in our dataset. To the best of our knowledge, ours is the most comprehensive illusion dataset available. We respectfully invite you to review Section 3.1 for details on filtering and our Limitation section for a detailed explanation of the challenges involved in expanding the dataset further.
>
> **Concern 3: *“The study does not discuss whether the illusions are representative of real-world scenarios, which would be vital for practical applications of VLMs. “***\
> We believe that certain categories such as real scenes, edited scenes, deceptive design, and hidden illusions are particularly relevant to real-world applications of VLMs. The ability to comprehend challenging but real image compositions and deceptive designs is crucial for VLM applications in robotics, as we highlight in Section 6.1.

---

> ### Author Response · Authors · 2024-06-06
> **Follow up with Reviewer 9wgj**
>
> Dear Reviewer 9wgj,
>
> We would love to hear back from you and make sure that all your concerns are addressed.
>
> Sincerely,
> Authors

---

### Official Review · Reviewer_XYvp · 2024-05-24

**Rating:** 7
**Confidence:** 4
**Ethics Flag:** 1

**Summary:**

This paper introduces IllusionVQA, a dataset for testing the ability of VLMs to locate and comprehend challenging optical illusions. The dataset consists of two tasks IllusionVQA-Comprehension and IllusionVQA-Soft-Localization. It is an interesting and novel problem and the community will be benefited from this perspective.

**Reasons To Accept:**

1. The dataset includes a diverse set of 12 distinct categories of optical illusions.

2. The proposed problem is relatively hard for current models. The authors evaluate a wide range of open-source and closed-source VLMs. Experiments show that VLMs can locate ordinary objects accurately but struggle with optical illusions.

3. The paper introduces a novel "soft localization" task that tests VLMs' ability to differentiate geometrically impossible objects from ordinary objects, which has not been explored in previous studies.

**Reasons To Reject:**

1. Potential to scale up the dataset is limited. The dataset size is relatively small, but it is totally understoodable, due to the difficulty to find additional high-quality optical illusions that met the inclusion criteria, which limits the scale of the dataset. Although the authors mentions synthetic optical illusion generation could be a potential way, but current image generation models have limited capabilities to follow such instructions.

---

> ### Author Rebuttal · Authors · 2024-05-30
>
> **Dear Reviewer XYvp**
>
> Thank you for appreciating our work on the “Soft Localization” benchmark which probes the VLM’s understanding of 3D geometry instead of commonsense reasoning or chart/graph understanding covered in previous studies.
>
> **Concern 1: “Potential to scale up the dataset is limited.”**\
> We agree that scaling up the IllusionVQA-Comprehension dataset is challenging due to the challenges mentioned in our Limitations section. Still, there is room for some enlargement using image manipulation such as image flipping, masking, color editing, etc but at the cost of dataset diversity. GVIL [1] expanded 16 root images to 100 to robustly test human-VLM agreement and HallusionBench [2] expanded 72 images to 144 to probe parametric memorization in VLMs. Adapting IllusionVQA-Comprehension images to use cases where diversity is less of a concern is a possible direction for future research.
>
> [1] Zhang et. al. - Grounding Visual Illusions in Language: Do Vision-Language Models Perceive Illusions Like Humans? [EMNLP 2023]\
> [2] Liu et al. - HallusionBench: An Advanced Diagnostic Suite for Entangled Language Hallucination & Visual Illusion in Large Vision-Language Models [CVPR 2024]

---

> > ### Comment · Reviewer_XYvp · 2024-06-01
> >
> > Thank you for addressing the comments and questions.

---

### Decision · Program_Chairs · 2024-07-10

**Decision:**

Accept

**Comment:**

The authors propose a new multiple choice VQA task, IllusionVQA, which
challenges models in understanding unusual images of various
categories.  A wide range of models perform poorly compared to humans
in both "explain why the image is unusual multi-choice"
("comprehension") and "pick which of these two objects is unusual"
("soft localization") settings. They will release their curated corpus.

Weaknesses included:

- Dataset size (374 images, XYvp: I will note that it would have been
  nice for the authors to include some notion of variance in their
  accuracy measures)

- a potential bias because GPT-4 was used during data curation (L4KJ);

- similarity to hallusionbench (7WKq);

- details of the dataset collection could have been expanded (9wgj);

I read over the responses the authors raised when addressing these
shortcomings, and am convinced by the responses.

Overall, the case for accepting IllusionVQA is clear: a task trivial
for the sighted human annotators, on which machines perform poorly, is
a valuable benchmark for the community to examine in more detail. And,
the fact that this dataset is split into various illusion categories
(compared to prior work) provides additional capacity for error
analyses in the future (e.g., it appears that some categories of
illusions are easier/harder than others for models).

[At least one review was discounted during the decision process due to quality]